# A Survey-Based Analysis of Injuries to Horses Associated with Transport by Road in New Zealand

**DOI:** 10.3390/ani12030259

**Published:** 2022-01-21

**Authors:** Christopher B. Riley, Chris W. Rogers, Kirrilly R. Thompson, Danielle Guiver, Barbara Padalino

**Affiliations:** 1School of Veterinary Science, Massey University, Palmerston North 4470, New Zealand; C.W.Rogers@massey.ac.nz (C.W.R.); Danielle.guiver@gmail.com (D.G.); 2School of Agriculture and Environment, Massey University, Palmerston North 4470, New Zealand; 3School of Medicine and Public Health, University of Newcastle, Newcastle, NSW 2308, Australia; Kirrilly.Thompson@newcastle.edu.au; 4Dipartimento di Scienze e Tecnologie Agro-Alimentari, Alma Mater Studiorum—Università di Bologna, Viale Fanin 50, 40126 Bologna, Italy; barbara.padalino@unibo.it

**Keywords:** horse, equine, road, transport, transportation, injury, survey, animal welfare

## Abstract

**Simple Summary:**

The road transport of horses poses a significant welfare issue. This study describes the injuries sustained by horses during road transport in New Zealand in general alongside factors associated with injury while in a moving vehicle. More than 1100 New Zealand horse industry participants were surveyed on their horse transport experiences and equine industry activities. Of the survey participants, approximately a fifth reported at least one horse injured during road transport during the two years covered by the survey. Most injuries (81%) occurred in transit when transported with one (39%) or more than one (21%) other horse. Most commonly, the hindlimbs, the head, or the forelimbs were injured (59% of horses). Injuries ranged from bruises to severe trauma. Factors associated with injury included horses used for eventing, not always checking the fitness of horses for transport, the use of a tail guard or bandage, a stallion guard in the vehicle, loose bedding on the floor, and behavioural problems. Overall, this survey identified a significant number of injuries and related euthanasia in horses transported by road in New Zealand and key factors associated with these injuries.

**Abstract:**

Negative outcomes associated with the road transport of horses are a significant welfare issue. This study aimed to describe the injuries sustained by horses during road transport in New Zealand and factors associated with trauma while in transit. New Zealand horse industry participants were surveyed on their horse transport experiences and equine industry involvement. Participants were solicited through horse organisations. The data were tabulated, and a logistic regression was performed to identify significant (*p* < 0.05) factors associated with transport-related injury. In total, 201/1133 (17.7%/2 years) eligible surveys reported at least one horse injured during road transport. Most incidents occurred in transit (137/169; 81%), or when transported with one (76/193; 39.4%) or more (41/193; 21.2%) other horses. Most commonly, the hindlimbs, the head, or the forelimbs were injured (59.1%; 110/186 horses), ranging in severity from bruises to catastrophic orthopaedic trauma necessitating euthanasia. Eventing, not always checking horses’ fitness for transport, using a tail guard or bandage, a stallion guard in the vehicle, bedding type on the floor, and behavioural problems were associated with injuries. This survey identified a significant incidence of injury and related death when horses are transported by road in New Zealand, and the key risk factors associated with the odds of injury.

## 1. Introduction

Stressful stimuli, such as those associated with road transport, can lead to homeostasis disruption with direct and indirect effects on the health status and physical performance of horses [1]. These stressors may be influenced by factors associated with the reasons for which transport is deemed necessary, such as competition, breeding, pleasure activities, sales, and slaughter. Furthermore, the activities or phases associated with road travel, including handling, loading, vehicle motion, and unloading, may affect equine physiology and behaviour in different ways [2,3].

Research into the welfare consequences of road transport for horses has expanded from studies primarily focused on their transport to slaughter [4,5,6] to those seeking to improve our understanding of the transport of equids for other broad commercial and non-commercial reasons [7,8,9,10,11,12,13]. Adverse outcomes associated with non-commercial transport are significant welfare issues yet to be addressed internationally in terms of transport practice, regulatory standards for vehicle operators of non-commercial vehicles, welfare-oriented vehicle design, and the appropriate training of all of those involved in horse transport [10,14].

The predominant adverse welfare outcomes associated with the road transport of horses have been characterised as behavioural [12,13,15,16] and physical (trauma) [6,9,11,13]. These two welfare issues are not independent. A strong association between perceptions of abnormal behaviour and the risk of injury for horses transported by road has been identified [12,13,17]. The injuries sustained by horses are a significant cause of morbidity, mortality, and economic loss [18], but published epidemiologic investigations of risk factors associated with these injuries are few [19]. Nevertheless, one Swedish study has identified the horse trailer (float) as the second most common location for an injury to a horse to occur, second only to those sustained in the paddock [20].

The New Zealand horse population is modest, with fewer than 100,000 horses nationally [21]. However, on a per capita basis, New Zealand has an estimated 20 horses per 1000 head of population, predominantly located around major human population centres [7]. This population compares to countries such as Switzerland, with a similar number of horses but a lower per capita base of 12 horses per head [22]. As racetracks, breeding facilities, and equestrian events are widely distributed throughout New Zealand, horses are frequently transported by road [6,7]. Based on this information, the authors were concerned that the exposure to the adverse consequences of road transport to horses might be significant. The associations between road transport management practices, journey characteristics, driver demographics and behaviour, and horse-related factors [5,6,9,10,11,13] have been described in recent studies of transport-related injury outside New Zealand. However, no evidence-based review of the injury risks to horses transported by road in this geographically distinct country has been published. In companion papers to the current manuscript, the authors have reported transport-related behavioural problems and identified human factors associated with road transport injury in New Zealand [17,23]. The present article describes the nature, extent, and impact of the injuries sustained by horses during road transport in New Zealand and identifies transport-related factors associated with trauma while in transit. This survey identified a significant annual risk of injury and death when horses are transported by road in New Zealand and several risk factors associated with the odds of injury.

## 2. Materials and Methods

This online survey was registered with the Massey University Human Ethics Committee as low risk (Ethics Notification Number: 4000017178). The overall methodology was mixed. Qualitative and quantitative data were collected via a self-report online survey. Descriptive statistics were used to determine the types of injuries sustained by horses during road transport in New Zealand, while inferential statistics enabled the identification of associated factors.

### 2.1. Respondents and Sample Size Estimate

New Zealand residents associated with the road transport of horses in New Zealand were the target population for the survey. Study participants were required to be ≥ 16 years old, to have at least one horse in their care, and to have organised or provided transport for one or more horses during the two years before completing the survey. The number of completed surveys required to obtain a 95% confidence interval with an error level of ± 3% was 1055, based upon an estimated target population of 90,000 equine industry participants [24,25].

### 2.2. Survey

The survey was constructed through an iterative design and review process to ensure valid questionnaire results [26,27] and using commercial software to capture information on a range of factors associated with the road transport of horses in New Zealand (Qualtrics, New Zealand). Open and closed-ended questions were designed to obtain respondents’ demographic details, information on their involvement with the equine industry, professional or amateur involvement with horses, the specific type of activities they participated in, horse experience, transport-related practices, and horse and road journey details. In addition to soliciting the information described above, survey respondents were queried on the most recent transport-related injury to a horse they were responsible for during the two years before completing the survey. To provide a comparison group, those who did not have a horse injured within this interval were asked questions regarding their most recent horse transport event during the same period. The survey questionnaire has previously been published by the authors elsewhere [17,23].

Invitations to participate and distribute or access the link to the survey were disseminated through New Zealand professional and amateur horse organisations and their members. Organisations were initially contacted by telephone with a reminder call six weeks after opening the survey. Industry organisation contacts who elected to participate were encouraged to forward the survey link to members and other contacts via email, and to promote the link via their website and social media platforms such as Facebook. Further distribution occurred courtesy of supportive individuals in a social media version of ‘snowball sampling’ [17,23,28]. The survey link was open for three consecutive months (7 February 2017 to 16 May 2017).

### 2.3. Data Analysis

Survey data concerning the human factors associated with transport-related injury in horses have been discussed elsewhere [23]. The current report focuses on non-human factors related to the risk of horse injury during road transport. Variable frequencies and summative statistics were calculated in Excel (Microsoft Excel, Version 16.46, 2021, Microsoft Corporation, Redmond, WA, USA). The mean and standard deviation were calculated for continuous variables normally distributed; the median and quartile ranges were calculated for those not normally distributed. All other statistical analyses were performed using R version 4.0.3 (R Foundation for Statistical Computing, Vienna, Austria). Logistic regression models were built using the ‘glm’ function from the base ‘stats’ package in R with injury (yes [horse injured]/no [horse not injured]) as the binary response variable. As most injuries occurred in transit (137/169; 81.1% of horses where the timing of when the injury occurred relative to the transport journey was reported), a univariate logistic regression was performed to identify factors associated with injury in transit. Statistical significance was determined (*p*-values) using the Wald test. Variables with a *p*-value <0.25 were considered in a multivariable model for injuries sustained in transit. Variables were progressively eliminated using a stepwise backward elimination procedure until all variables in the final model had a *p*-value < 0.05. The outcomes of analyses were reported as odds ratios (OR) with 95% confidence intervals for each variable.

## 3. Results

### 3.1. Survey Response Level

Of the 1486 people who commenced the online survey, 1133 (76.2%) met the inclusion criteria of having transported a horse by road within the two years before the survey date. This sample resulted in the targeted 95% confidence level and an error level of ±3% for the study population. The data describing the respondents’ demographics, education, training, and experience (human factors) have been previously published [23].

### 3.2. Descriptive Data for Road Transport-Related Horse Injuries

Two hundred and four respondents reported 205 incidents in which at least one horse was injured in association with road transport. Of these, 201 horse injuries had occurred within the two years before the survey (2015–2017), resulting in an injury incidence over two years of 17.7% (i.e., average 8.9%/year); those that occurred before this time were excluded from further analysis. Of the 201 respondents, 192 provided details of 193 incidents and the injuries sustained. This included 189 single events involving injury to one horse, two separate events that occurred with the same horse, and two events that resulted in injury to three horses on each occasion (i.e., injury to 196 horses) (Figure 1). The breed and sex of the injured horses are summarised in Table 1.

Incidents were reported to have occurred during pre-loading (1/193; 0.5%), loading (19/193; 9.8%), during the journey ((137/193; 71%), including injuries at the beginning (26/193; 13.5%), middle (90/193; 46.6%), or end (21/193; 10.9%) of the journey), or during unloading (12/193; 6.2%). The remaining respondents did not know at what stage of the journey the injury had occurred (23/193; 11.9%) or did not enter a response (1/193; 0.5%). Of the 169 incidents where the timing of the event resulting in injury was known, 18.9% (23/169) occurred during pre-loading/loading/unloading, and 81.1% (137/169) occurred in transit. Of 193 incidents, most horses were transported with one (76/193; 39.4%) or more (41/193; 21.2%) accompanying horses; 59/193 (30.6%) were transported singly. For 11/193 (5.7%), it was unclear if the horse was transported singly or with others (transported by a commercial trucking company, a veterinarian, or a person other than the respondent). Six respondents did not answer this question. The most common single reason that respondents thought an injury had occurred was a behavioural or physical problem (e.g., ‘caught boot on hinges of the door to the inside of the truck and panicked and fell over’). Multiple or other factors, including some human-related factors, were also reported (Table 2).

Anatomic patterns of injury (sites of trauma) described by respondents (where provided) are summarised in Table 3. A single affected region was most commonly described (59.1%; 110/186 horses). Injury affecting a single area was most common for the hind limbs, the head and muzzle, and the forelimbs. Injuries involving multiple body sites were reported for 40.9% (76/186) of these horses (Table 3).

Concerning the type of injury and tissue trauma, 190/201 respondents provided further information (Table 4). Reported injuries ranged from bruises to catastrophic orthopaedic injury, including fractures and joint trauma. Most horses (89.5%; 170/190) were reported as having recovered at the time of the survey, 7.4% (14.190) had not recovered, and 3.2% (6/190) were euthanised due to their injuries. Of the recovered horses, 43.5% (80/170) did so within a week, 21.2% (36/170) required 1 to 2 weeks, 17.1% (29/170) required 2 to 4 weeks, 5.3% (9/170) required 2 to 4 weeks, and 9.4% (16/170) required more than 2 months.

The respondents most frequently reported treating the injured horse themselves (85/188; 45.2%). Veterinary treatment was provided for 25.5% (48/188) of horses at the location where the injury was found, 8.5% (16/188) were treated by a veterinarian at a clinic or hospital, and respondents treated 5.9% (11/188) with the assistance of a friend. The remainder (28/188; 14.9%) did not provide treatment for their horse. Where the cost of treating the injured horse was reported (81.9%; 158/193), NZD 0 was indicated by 36.7% (58/158) of respondents, and a median of NZD 300 (IQR NZD 100–1000; range NZD 5–12,000) by those who paid for treatment (100/158). The costs of vehicle repairs, time lost from performance activities or a use change, and time caring for injured horses were not reported.

### 3.3. Variables Associated with Injury

The variables and the frequency of categories within each variable explored via a logistic regression are listed in Table 5. Univariate screening of the variables presented in Table 6 identified the following as inclusion variables for the multivariate model: horse industry sector (i.e., eventing), not always checking the fitness of horses before transport, the use of a tail guard or bandage on the transported horse, the presence of a stallion guard (head partition) in the transport vehicle, the bedding type used on the floor of the transport vehicle, and reporting the horse as having transport-related behavioural problems. The results for variables that failed to meet inclusion criteria are listed in Appendix A. In the final multivariable logistic regression model, horse industry sector (eventing), the average number of horses transported, not always checking the fitness of horses for travel before transport, the presence of a stallion guard (head partition) in the transport vehicle, and reporting the horse as having transport-related behavioural problems were associated with an increase in the odds of injury in transit. In contrast, a bare floor or rubber matt used on the floor of the transport vehicle was associated with a decrease in odds (Table 7).

## 4. Discussion

### 4.1. Descriptive Data for Road Transport-Related Horse Injuries

The horse level incidence of injury for New Zealand respondents’ horses (8.9%) is comparable with a yearly average of 4.5% reported in a smaller sample of non-commercial interviewees in an Australian study [9], and with 11.5% by Italian respondents [13], but significantly lower than the 45% recorded for Australian respondents [11] to similarly designed online surveys. The overall rate of injury found in a United Kingdom survey of non-commercial horse transport was similar (8.1%), but the timeframe covered was not stated in this work [29]. In comparison, injury rates reported for commercial long-haul road shipment in Australia are much lower (0.18%/year) [30].

The contemporary rates of injury reported for horses transported to slaughter are higher. A report on Mexican and USA horses transported to slaughter to a Canadian plant described an injury rate of 11% over four months [5]. Slaughter horses unhabituated to road transport have high injury rates, with 58% of adults and 17% of foals injured following short journeys to an Icelandic meat plant during a ten-week study period [6]. However, many horses transported for commercial reasons such as slaughter are unlikely to be trained or experienced in the modes of road transport employed and may not present an ideal comparison population for horses in the current study [6].

The transport-related incidence of injuries resulting in euthanasia in the current study (3.2%) approximates the 3.3% deaths/euthanasia reported for the Australian industry [31], and is only slightly higher than the 2% of deaths reported for non-commercial transport incidents in the United Kingdom [29]. These transport-related findings emphasise the need to address the high degree of welfare concern for horses transported by road for non-commercial purposes.

In this survey, injuries were reported to occur during all stages of transport: preparation for loading (pre-loading), loading, unloading, and in transit, but most occurred in transit (81%). The higher frequency of in-transit injuries is comparable to the value reported in an Australian study (84%), where methods and vehicles used for the non-commercial road transport of horses are comparable [9] but higher than the frequency reported in a recent survey of Italian horse industry participants (65%) [13]. The United Kingdom survey of transport-related injuries only described transit incidents and near misses [29]. Interestingly in the current study, most injuries were sustained when horses were transported with companions (60.8%), but injuries associated with kicks, bites, or negative interaction with another horse were reported in only 18.8%. It is unclear if evasion or other behaviours may have contributed to the injury rate in horses transported with others, but the findings contrast with the recent United Kingdom study that found most horses were injured during journeys (63.9%) while travelling alone [29].

Anatomic patterns of transport-related injury (sites of trauma) described by respondents may provide insight into their causes and possible prevention. Still, few reports have provided these details in depth [9,23,29]. In New Zealand, horses most commonly suffered injuries at multiple concurrent sites (59.1%), a more frequent occurrence than that reported in contemporary studies conducted in Italy (47.1%) [13] and the United Kingdom (20.9%) [29]. The reason for this difference is not clearly identified. However, the general terrain of New Zealand’s islands is highly mountainous, and in many regions, roadways between population centres are often winding with tight, sharp turns and steep gradients encountered during road transport. The efforts spent towards balance preservation during cornering and in response to road inclines are associated with equine stress and may influence the rates of injury that are related to scrambling and slipping [32,33].

Injuries most commonly affected the hindlimbs, singly and in combination with other locations of trauma (Table 3). In agreement with an Australian study, many of these were associated with loss of balance or falling, slipping or scrambling, or a combination of these factors [9] (Table 2). Slipping and scrambling are frequently considered behavioural problems by owners [9,12]. However, it is more likely that these are associated with horses that cannot compensate for the increase in angular momentum of the hindquarters when turning or braking at speed, particularly when travelling in a forward-facing direction [32,34]. Unlike the forelimbs, a base-wide stance in the hindlimbs is not achievable for most horses under these conditions, therefore they crouch on their hindquarters, increasing the risk of losing traction and slipping or scrambling. More than 20% of transport injuries described by Hall et al. (2020) also affected the hindlimbs; however, although 65% of the vehicles described in the United Kingdom were trailers, the authors did not report the direction of travel.

Based upon univariate and multivariate analyses in the current study, the type of flooring or bedding also correated with the risk of injury, possibly due to the loss of hoof traction during transport manoeuvres. A lack of rubber matting was associated with an increased risk of injury in a recent Italian study of road transport [9]. Interestingly, although leg bandages or boots for travel are popular among horse owners to protect the limbs, they neither decreased nor increased the odds of an injury in this study of New Zealand horses. Therefore, in agreement with a previous report, their popular perception as protection from harm during transport is not supported by evidence [9].

The second most common location of injury was to the head, muzzle, and neck. This was a more frequent finding in the New Zealand population (34.9%) than reported in Italy (11.8%) [13] and the United Kingdom (5.8%) [29]. Such injuries reportedly occur more frequently in forward-facing horses [34]; the direction of travel most commonly reported in the current study (74.1%). The significant increase in the odds of injury associated with stallion guards (×1.6), a design feature associated with floats (trailers) to prevent oral horse-to-horse conflict during forward-facing transport, suggests that removing or redesigning this feature may decrease the frequency of this type of head injury. However, some of the head or neck injuries reported by respondents were attributed to social conflict with another horse, suggesting that care should be taken to transport socially compatible herd mates. Interestingly, univariate analyses indicate a trend for poll protectors to be associated with increased odds of injury; this device does not appear to be protective against head injury [9].

Whilst respondents reported injuries ranging from bruises to catastrophic orthopaedic trauma necessitating euthanasia, shallow cuts or wounds were most common (62.1%). These may arise from contact with the transport vehicle or the horse stepping on itself, as such injuries are commonly associated with scrambling or slipping. Respondents likely underestimated the frequency of bruising associated with such transport injuries because of a lack of outwardly visible signs [35]. Most injuries were perceived as minor. This perception may explain the limited involvement of veterinarians in assessment and treatment. It is likely that veterinary treatment was sought only for injuries perceived as complex or severe by respondents (see Table 4). Our data support this hypothesis, with the percentage of injuries where veterinary treatment was immediately sought (25.5%) being similar to the percentage of injuries that respondents reported as deep wounds or fractures (Table 4). Most respondents treated equine transport-related injuries without veterinary assistance or did not treat them. This finding is consistent with a previous cross-sectional survey of equine wounds in New Zealand that found 58% of horse owners did not seek professional veterinary assistance for injured horses [18]. The lack of veterinary consultation and treatment of injured horses is a possible welfare concern, as owners often fail to observe or recognise the significance of health conditions affecting their horses [36].

For reasons relating to the survey’s design (i.e., the time required for completion), the investigation of the economic consequences of transport-related equine injuries was limited to the cost of treatment. These costs were consistent with those reported for wound management in New Zealand in the aforementioned cross-sectional survey [18].

### 4.2. Variables Associated with Injury

The differences in study population demographics, geography, and other risk factors may contribute to a better understanding of similar or dissimilar findings of the comparable studies cited above [13,29]. For this reason, the authors of the current and other published works in the field have used regression methods to identify possible risk factors and gauge the statistical significance of these associations [9,11,13]. Others have recently adopted this approach [29].

The odds of injury in the horse were significantly lower if the horse was always checked for fitness before transport. Failure to carefully check fitness for the return journey may be a result of horse handlers being fatigued themselves after engaging in performance activities [9]. Supporting this possibility, an Australian study found an exacerbation of muscular problems in horses when they were not assessed before a journey [30].

The industry sector (i.e., eventing) was associated with transport-related injury, in agreement with others, although the method used for industry classification differed from other published works [13,29]. Eventers are at a high risk of orthopaedic injuries associated with training and performance activities [37,38,39]. Some eventing horses may have sustained injuries not reported or detected by owners, increasing their risk of scrambling or slipping during transport. A limitation of the current study is that the survey did not identify if an injury occurred on an outward or inward journey, something that should be discerned in future surveys.

Equipment such as poll protectors, leg bandages, tail bandages or bags, and blankets are often used to protect horses during transport. However, evidence of their efficacy in preventing injury is lacking. For example, in the current study, using a tail bandage or guard as a protective device was associated with increased odds of injury. For horses transported in Italy, in general, horse protective equipment and vehicle features designed to protect horses were associated with increased odds of problem-related behaviours, and therefore with injury [13]. This suggests that people who anticipate behavioural difficulties during transport may use protective equipment to mitigate the risk of injury. The authors suggest that the retraining of these horses may be a more effective risk mitigation strategy [17].

In New Zealand, trailer or truck safety design features were not associated with increased or decreased odds of injury. These findings are counterintuitive and difficult to explain. However, little research has been conducted into transport vehicle designs that prioritise equine welfare over cost and human convenience [14,40]. Shavings and sawdust are often used to cover the floor in non-commercial vehicles to absorb faecal moisture and urine; much like the same materials are used in stables. In the current study, the use of shavings or sawdust was associated with an increased odds of injury compared to rubber matting; others have found an increased association with rubber matting [13]. Surfaces that are loose or slippery are more likely to cause difficulty in maintaining footing when cornering or braking. Flooring is a vehicle design feature that requires further objective scrutiny [14].

Unsurprisingly, a strong association between injury and behavioural problems as perceived by respondents was confirmed, a finding that is in agreement with previously published works of the authors and is discussed extensively elsewhere [12,13,17]. Further research should focus on the definition of behavioural problems that are transport-specific, including differentiating those that are responses to driving practices from those that are associated with maladaptation to the features of the transport vehicle environment [9]. This will enable suitable recommendations for foundation training of both horses and drivers for transport.

## 5. Conclusions

The results of this representative survey found a high annual incidence of injury at the horse level and a concerning number of deaths associated with the road transport of horses in New Zealand. These works contribute to the growing recognition that the road transport of horses poses a significant welfare concern that requires a concerted response by the equine industry, commercial and non-commercial. Studies such as ours can identify specific areas for targeted prospective research and intervention. However, injuries to horses associated with transport by road should always be considered within a broader context of driver education and well-being and horse training, management, handling, and welfare.

## Figures and Tables

**Figure 1 animals-12-00259-f001:**
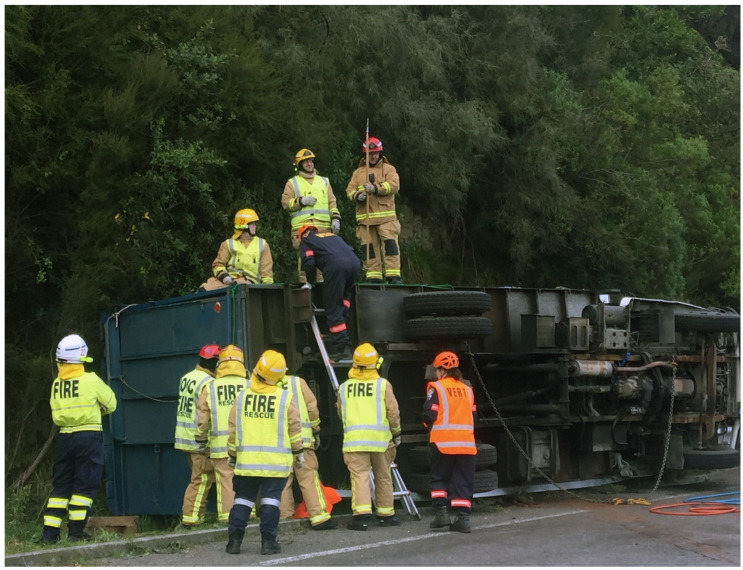
An incident involving the rollover of a truck containing three horses. The horses sustained minor wounds to the limbs.

**Table 1 animals-12-00259-t001:** The breed and sex of injured horses as described by respondents to a survey on road transport-related equine injury in New Zealand. Data were available for 196/204 injured horses involved in 193 incidents.

Breed	Mare/Filly	Gelding	Stallion	N (%)
Arabian (including Anglo Arabs and part Arab)	6	6	0	12 (6.1)
Appaloosa (including two crossbred)	2	2	0	4 (2.0)
Draught (including five crossbred)	1	6	0	7 (3.6)
Pony (including three crossbred)	5	4	0	9 (4.6)
Quarter horses (including two crossbred)	2	4	0	6 (3.1)
Standardbred	7	8	0	15 (7.7)
New Zealand Stationbred	5	6	0	11 (5.6)
Thoroughbred ^1^	34	33	2	70 (35.7)
Thoroughbred crosses	8	9	0	17 (8.7)
Warmblood (including six crossbred)	14	7	0	21 (10.7)
Other breeds and crossbreeds	11	10	0	21 (10.7)
Not described	1	1	1	3 (1.5)
Total (%)	95 (48.5)	97 (49.5)	4 (2.0)	196 ^2^

^1^ The sex of one thoroughbred was not provided; ^2^ One horse was injured during two separate incidents; therefore, breed and sex for this horse are listed only once in the table.

**Table 2 animals-12-00259-t002:** Factors reported by respondents as associated with the injury of horses during road transport in New Zealand. Respondents provided details for 192/193 (99%) incidents.

Factor ^1^	Number Listing This Factor	Details and Examples	Number Listing This Detail
A mistake by the driver	18		
		Excess speed.	5
		Fatigue.	4
		Other, e.g., failure to correctly couple the float to the vehicle.	9
A problem with the horse	127		
		Loss of balance or falling.	28
		Slipping or scrambling.	38
		Horse kicked, bitten, or had negative interaction with another horse.	25
		Horse jumped or fell out of vehicle.	5
		Vocalisation.	2
		Other, e.g., failure to untie the head when unloading; loading problems; reaction to a passing vehicle; anxiety.	24
		Unsure.	5
Poor road conditions	7		
A mechanical problem with the transport vehicle	7		
Vehicle collision or accident	12		
		Collision with another vehicle.	7
		Other, e.g., an animal on the road; truck rolled; heavy braking.	5
Other factors	27		

^1^ 181/192 (94.8%) respondents reported a single contributing factor; 10/192 (5.2%) reported multiple contributing factors.

**Table 3 animals-12-00259-t003:** The anatomic distribution of injuries as described by 186/201 respondents reporting a horse injured during road transport in New Zealand.

Injury Combinations and Anatomic Regions Affected ^1^								
	Head/Muzzle	Neck	Thorax	Hindquarters or Abdomen	Forelimbs	Hind Limbs	Tail Head/Tail	N (%)
	⊠	-	-	-	-	-	-	20 (10.8)
	⊠	⊠	-	-	-	-	-	2 (1.1)
	⊠	⊠	-	-	-	-	⊠	1 (0.5)
	⊠	⊠	⊠	-	⊠	-	-	1 (0.5)
	⊠	⊠	-	⊠	-	⊠	-	1 (0.5)
	⊠	⊠	-	-	⊠	⊠	-	1 (0.5)
	⊠	⊠	⊠	⊠	⊠	⊠	⊠	1 (0.5)
	⊠	-	⊠	⊠	⊠	⊠	-	2 (1.1)
	⊠	-	⊠	⊠	⊠	⊠	⊠	2 (1.1)
	⊠	-	⊠	-	⊠	⊠	-	1 (0.5)
	⊠	-	-	⊠	-	-	-	1 (0.5)
	⊠	-	-	⊠	-	⊠	-	2 (1.1)
	⊠	-	-	⊠	-	-	⊠	1 (0.5)
	⊠	-	-	⊠	-	⊠	⊠	1 (0.5)
	⊠			⊠	⊠	⊠		2 (1.1)
	⊠	-	-	-	⊠	-	-	1 (0.5)
	⊠	-	-	-	⊠	⊠	-	2 (1.1)
	⊠	-	-	-	⊠	⊠	⊠	1 (0.5)
	⊠	-	-	-	-	⊠	-	4 (2.2)
	⊠	-	-	-	-	⊠	⊠	1 (0.5)
	⊠	-	-	-	-	-	⊠	1 (0.5)
	-	⊠						1 (0.5)
	-	⊠	⊠	-	-	-	-	2 (1.1)
	-	⊠	⊠	⊠	-	-	-	1 (0.5)
	-	⊠	-	-	⊠	-	-	1 (0.5)
	-	⊠	-	-	⊠	⊠	-	1 (0.5)
	-	⊠	-	⊠	⊠	⊠	⊠	1 (0.5)
	-	⊠	-	-	-	⊠	⊠	1 (0.5)
	-	⊠	-	-	-	-	⊠	1 (0.5)
	-	-	⊠	⊠	-	-	-	1 (0.5)
	-	-	⊠	⊠	-	-	⊠	1 (0.5)
	-	-	⊠	-	⊠	-	-	4 (2.2)
	-	-	⊠	-	-	⊠	-	3 (1.6)
	-	-	⊠	-	-	⊠	⊠	1 (0.5)
				⊠				5 (2.7)
	-	-	-	⊠	⊠	-	-	1 (0.5)
	-	-	-	⊠	-	⊠	-	5 (2.7)
	-	-	-	⊠	-	-	⊠	2 (1.1)
	-	-	-	⊠	⊠	⊠	-	2 (1.1)
	-	-	-	⊠	-	⊠	⊠	1 (0.5)
	-	-	-	⊠	⊠	⊠	⊠	2 (1.1)
	-	-	-	-	⊠	-	-	15 (8.1)
	-	-	-	-	⊠	⊠	-	6 (3.2)
	-	-	-	-	⊠	⊠	⊠	1 (0.5)
	-	-	-	-	⊠	-	⊠	1 (0.5)
	-	-	-	-	-	⊠	-	58 (31.2)
	-	-	-	-	-	⊠	⊠	8 (4.3)
	-	-	-	-	-	-	⊠	11 (5.9)
Total	49	16	20	35	49	111	40	186
(%)	(26.3)	(8.6)	(10.8)	(18.8)	(26.3)	(59.7)	(21.5)	(100)

⊠ = injury; ^1^ rows show injury combinations reported by respondents.

**Table 4 animals-12-00259-t004:** The nature and combinations of tissue damage described by 190/201 respondents reporting an injured horse during road transport in New Zealand.

Tissue Injury Types and Concurrent Combinations ^1^							
	Bruise	Skin or Tail Rubbed Raw	Shallow Cut or Wound	Deep Cut or Wound	Fractured Bone	Other Injuries ^2^	N (%)
	⊠	-	-	-	-	-	12 (6.3)
	⊠	⊠	-	-	-	-	1 (0.5)
	⊠	-	⊠	-	-	-	27(14.2)
	⊠			⊠			4 (2.1)
	⊠	-	-	-	-	⊠	5 (2.6)
	⊠	⊠	⊠	-	-	-	5 (2.6)
	⊠	⊠	-	⊠	-		1 (0.5)
	⊠	⊠	-	-	-	⊠	1 (0.5)
	⊠	-	⊠	⊠	-	-	3 (1.6)
	⊠	-	⊠	-	-	⊠	2 (1.1)
	⊠	⊠	⊠	⊠	-	-	1 (0.5)
	⊠	⊠	⊠	-	-	⊠	1 (0.5)
	⊠	⊠	-	⊠	-	⊠	1 (0.5)
	⊠	-	⊠	⊠		⊠	1 (0.5)
	⊠	-	⊠	⊠	⊠	-	1 (0.5)
	⊠	⊠	⊠	⊠		⊠	1 (0.5)
	-	⊠	-	-	-	-	9 (4.7)
	-	⊠	⊠	-	-	-	12 (6.3)
	-	⊠	⊠	-	-	⊠	2 (1.1)
	-	⊠	-	⊠	-	-	1 (0.5)
	-	-	⊠	-	-	-	57(30.0)
	-	-	⊠	⊠	-	-	1 (0.5)
	-	-	⊠	-	⊠	-	1 (0.5)
	-	-	⊠	-	-	⊠	2 (1.1)
	-	-	⊠	⊠	⊠	-	1 (0.5)
	-	-	-	⊠	-	-	20(10.5)
	-	-	-	⊠	⊠	-	2 (1.1)
	-	-	-	⊠	-	⊠	2 (1.1)
	-	-	-	-	⊠		5 (2.6)
	-	-	-	-	-	⊠	8 (4.2)
Total	67	36	118	40	10	67	190
(%)	(35.3)	(18.9)	(62.1)	(21.1)	(5.3)	(35.3)	(100)

⊠ = injury; ^1^ rows show combinations of tissue injury reported by respondents. ^2^ Other injuries described included: muscle injury (7); shoes ripped off hooves (5); joint damage (4); eye injury (3); extreme fatigue (1); suspensory injury (1); capped hocks (1); sacroiliac injury (1); tendon injury (1); severe dehydration and distress (1); colic and shock (1).

**Table 5 animals-12-00259-t005:** Frequency table of the in-transit associated responses to a survey on horse road transport injuries in New Zealand.

Variable Name	Category	Count (%)
Horse industry sector	Breeding	63 (5.9)
	Dressage	122 (11.4)
	Endurance and competitive trail riding	48 (4.5)
	Eventing	110 (10.3)
	Pony club	64 (6.0)
	Recreational riding	195 (18.2)
	Showjumping	133 (12.4)
	Showing	45 (4.2)
	Standardbred racing	61 (5.7)
	Thoroughbred racing	168 (15.7)
	Other	64 (6.0)
	Total	1073 (100)
Number of horses normally transported ^1^	Total	1047 (97.6)
	Missing values	26 (2.4)
Responder self-assessed ability to identify distress in horses	5—Very high	559 (52.1)
	4—High	441 (41.1)
	3—Moderate	60 (5.6)
	2—Some	11 (1.0)
	1—None	0 (0)
	Total	1171 (99.8)
	Missing values	2 (0.2)
Journey frequency	Daily	75 (7.0)
	Two to five times a week	268 (25.0)
	Once weekly	268 (25.0)
	Fortnightly	205 (19.1)
	Monthly	121 (11.3)
	Less than once a month	144 (13.4)
	Total	1073 (100)
Average journey length (km) ^2^	Total	1030 (96.0)
	Missing values	43 (4.0)
Horse checked for fitness to travel	Always	691 (64.4)
	Not always	380 (35.4)
	Total	1071 (99.8)
	Missing values	2 (0.2)
Vehicle type	Float/trailer—angle load	119 (11.1)
	Float/trailer—straight load	555 (51.7)
	Small truck—two to three horses	125 (11.6)
	Large truck—more than three horses	205 (19.1)
	Commercial truck	69 (6.4)
	Total	1073 (100)
Direction of travel	Head facing or angled to the front	795 (74.1)
	Head facing or angled to the rear	238 (22.2)
	Horse free in the vehicle	6 (0.6)
	Total	1039 (96.8)
	Missing values	34 (3.2)
Sedation used for transport	No	804 (74.9)
	Yes	269 (25.1)
	Total	1073 (100)
Horse specifically trained for loading and travelling	No	227 (21.2)
	Yes	846 (78.8)
	Total	1073 (100)
Poll protector used	No	1041 (97.0)
	Yes	32 (3.0)
	Total	1073 (100)
Neck rug used	No	1015 (94.6)
	Yes	58 (5.4)
	Total	1073 (100)
Body rug used	No	748 (69.7)
	Yes	325 (30.3)
	Total	1073 (100)
Leg bandage/boots used	No	452 (42.1)
	Yes	621 (57.9)
	Total	1073 (100)
Tail guard/bandage used	No	631 (58.8)
	Yes	442 (41.2)
	Total	1073 (100)
Head restraint in the moving vehicle	Cross-tied	32 (3.0)
	Restrained on elastic bungee cords	70 (6.5)
	Tied on a long rope	222 (20.7)
	Tied on a short rope	618 (57.6)
	Not restrained	112 (10.4)
	Total	1054 (98.2)
	Missing values	19 (1.8)
Padded walls in the vehicle	No	571 (53.2)
	Yes	502 (46.8)
	Total	1073 (100)
Padded chest bar in the vehicle	No	546 (50.9)
	Yes	527 (49.1)
	Total	1073 (100)
Padded bum (rear) bar in the vehicle	No	665 (62.0)
	Yes	408 (38.0)
	Total	1073 (100)
Padded partition in the vehicle	No	512 (47.7)
	Yes	561 (52.3)
	Total	1073 (100)
Partition between horses extending to the floor	No	970 (90.4)
	Yes	103 (9.6)
	Total	1073 (100)
Stallion guard (head partition) in the vehicle	No	785 (73.2)
	Yes	288 (26.8)
	Total	1073 (100)
Bedding on the floor	Rubber mat	858 (80.0)
	Shavings or sawdust	102 (9.5)
	None (bare floor)	104 (9.7)
	Total	1064 (99.2)
	Missing values	9 (0.8)
Food available en route	No	728 (67.8)
	Yes	345 (32.2)
	Total	1073 (100)
Transport-related behavioural problems reported	No	834 (77.7)
	Yes	239 (22.3)
	Total	1073 (100)

^1^ Median of 5 horses (IQR 3–13; range 1–300); ^2^ median 50 km (IQR 30–100; range 1–3800).

**Table 6 animals-12-00259-t006:** Variables with a Wald test *p*-value < 0.25 selected for initial inclusion in the multivariate logistic regression model based on the results of univariate logistic regression analyses of associations between horse injury in transit and industry sector, vehicle safety design features, equipment used to protect horses in transit, and transport-related behavioural problems.

Variable Name	Category	Est. ^1^	SE ^2^	OR ^3^	95% CI ^4^	*p* ^5^
Horse industry sector	Breeding	−0.18	0.63	0.83	0.23–2.92	0.034
	Dressage	0.05	0.53	1.05	0.39–3.16	
	Endurance/competitive trail riding	−0.13	0.68	0.88	0.21–3.26	
	Eventing	1.10	0.49	2.99	1.23–8.44	
	Recreational riding	0.21	0.49	1.23	0.50–3.47	
	Showjumping	0.76	0.48	2.13	0.87–6.01	
	Showing	0.74	0.58	2.09	0.67–6.81	
	Standardbred racing	0.23	0.59	1.25	0.39–4.11	
	Thoroughbred racing	0.27	0.49	1.31	0.53–3.72	
	Other	0.80	0.54	2.23	0.81–6.80	
	Pony club	Ref				
Number of horses transported		0.006	0.003	1.00	1.00–1.01	0.064

Responder self-assessed ability to identify distress in horses	5—Very high	−1.75	1.42	0.17	0.007–4.44	0.166
	4—High	−1.93	1.42	0.15	0.006–3.71	
	3—Moderate	−2.2	1.49	0.11	0.004–3.05	
	2—Some	−1.56	4.39	0.00	0.00–0.00	
	1—None	Ref				
Journey frequency	Daily	0.53	0.52	1.69	0.59–4.79	0.107
	Two to five times a week	1.02	0.40	2.77	1.33–6.54	
	Once weekly	0.79	0.41	2.20	1.03–5.24	
	Fortnightly	0.96	0.41	2.61	1.22–6.28	
	Less than once a month	0.76	0.44	2.15	0.93–5.38	
	Monthly	Ref				
Average journey length (km)		0.0005	0.0003	1.00	1.00–1.00	0.087

Horse checked for fitness to travel	Always	0.50	0.18	1.65	2.34	0.006
	Not always	Ref				
Direction of travel	Head facing/angled to the front	−0.81	0.42	0.45	0.20–1.08	0.031
	Head facing/angled to the rear	−0.36	0.44	0.70	0.31–1.74	
	Horse free in the vehicle	Ref				
Poll protector used	Yes	0.78	0.42	2.18	0.90–4.76	0.081
	No	Ref				
Tail guard/bandage used	Yes	0.48	0.18	1.61	1.14–2.29	0.008
	No	Ref				
Stallion guard (head partition) in vehicle	Yes	0.45	0.19	1.56	1.07–2.26	0.020
	No	Ref				
Bedding on the floor	Shavings or sawdust	0.59	0.27	1.80	1.05–2.98	0.021
	None (bare floor)	−0.008	0.31	0.99	0.51–1.78	
	Rubber mat	Ref				
Food available en route	Yes	0.28	0.18	1.33	0.91–1.90	0.129
	No	Ref				
Transport-related behavioural problems	Yes	1.14	0.19	3.13	2.16–4.51	<0.001
	No	Ref				

^1^ Coefficient estimate; ^2^ standard error; ^3^ odds ratio; ^4^ 95% confidence interval; ^5^ Wald test *p*-value.

**Table 7 animals-12-00259-t007:** Multivariate logistic regression analyses of associations between horse injury in transit and industry sector, vehicle features, horse protective equipment used with horses in transit, and transport-related behavioural problems.

Variable Name	Category	Est. ^1^	SE ^2^	OR ^3^	95% CI ^4^	*p*^5^ |>z|	*p* ^6^
Horse industry sector	Breeding	−0.42	0.68	0.66	0.16–2.48	0.538	0.030
	Dressage	−0.07	0.54	0.93	0.33–2.85	0.891	
	Endurance or competitive trail riding	−0.07	0.70	0.93	0.22–3.61	0.922	
	Eventing	0.99	0.50	2.70	1.07–7.82	0.046	
	Recreational riding	0.18	0.50	1.12	0.48–3.45	0.714	
	Showjumping	0.62	0.50	1.87	0.74–5.40	0.210	
	Showing	0.68	0.60	1.97	0.61–6.65	0.257	
	Standardbred racing	−0.21	0.64	0.81	0.23–2.89	0.742	
	Thoroughbred racing	0.05	0.52	1.05	0.39–3.17	0.919	
	Other	0.86	0.55	2.36	0.82–7.44	0.120	
Number of horses transported		0.007	0.003	1.01	1.00–1.01		0.017
Horse not always checked for fitness to travel		0.45	0.19	1.56	1.07–2.27		0.015
Stallion guard (head partition) in vehicle		0.48	0.20	1.61	1.08–2.39		0.033
Bedding on the floor	Rubber mat	−1.78	0.81	0.17	0.04–0.95	0.030	0.035
	Shavings or sawdust	−1.14	0.84	0.32	0.06–1.90	0.184	
	None (bare floor)	−1.93	0.86	0.15	0.03–0.88	0.026	
Transport-related behavioural problems		1.19	0.20	3.27	2.22–4.82		<0.0001

^1^ Coefficient estimate; ^2^ standard error; ^3^ odds ratio; ^4^ 95% confidence interval; ^5^ Wald test *p*-value for category within variable; ^6^ Wald test *p*-value for variable.

## Data Availability

Data available on request due to restrictions (human ethics approval).

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
