# Peer review of "A Survey-Based Analysis of Injuries to Horses Associated with Transport by Road in New Zealand"

_animals, 2022, doi:10.3390/ani12030259_

Round 1
Reviewer 1 Report
The title accurately reflects the major findings of the work.
The abstract adequately summarizes methodology, results, and significance of the study.
Keywords represent the article adequately.
The introduction section is well written and it falls within the topic of the study, however, Authors should add more information and related references regarding the animal’s physiological response to a stressful condition as transportation. At the beginning of introduction I suggest to write “Under stressful stimuli as the body must find a new dynamic equilibrium and this requires several adaptive body responses. Transport represents a stressful stimulus that can lead to homeostasis disruption with direct effect on animal health status and physical performance of the animals (Fazio F. et al., Journal of Veterinary Behavior, 26, 5-10; Carcangiu V. et al., Archives Animal Breeding, 61 (1), 37-41). Horses are transported for different reasons, including competitions, breeding, pleasure activities, sales, and slaughtering. Travel includes handling, loading, transport in self, unloading, and often adaptation to a new environment; each of these phases affects horse physiology and behavior in a different way (Rizzo M., et al., Journal of Veterinary Behavior 18 (2017) 56-61; Rizzo M., et al., Journal of Equine Veterinary Science 48 (2017) 23–30; Rizzo M., et al., Journal of Equine Veterinary Science 55 (2017) 84–89).”
The section of Materials and Methods is clear for the reader and it meticulously describes the methods applied in the study.
Results section as well as Discussion section is clear and well written. The findings obtained in the study were well discussed and justified with appropriate references.
The conclusion section is well written and the main findings as well as the significances of the study are well reported.
The tables and Figure are generally good and well represent the results of the study.
Authors should check and standardize the references in the list according to journal guidelines.
Author Response
Dear Reviewer
Thank you for your feedback. We have addressed your feedback below and incorporated the changes into the revised manuscript.
An additional paragraph as suggested by the reviewer has been added, while retaining the equine transport focus and style of the manuscript.
“Stressful stimuli, such as those associated with road transport, can lead to homeostasis disruption with direct and indirect effects on the health status and physical performance of horses [1]. These stressors may be influenced by factors associated with the reasons for which transport is deemed necessary, including competition, breeding, pleasure activities, sales, and slaughter. Furthermore, the activities or phases associated with road travel including handling, loading, vehicle motion, and unloading, may affect equine physiology and behaviour in different ways [2,3].”
Ferlazzo, A.; Fazio, E.; Medica, P. Behavioral features and effects of transport procedures on endocrine variables of horses. J. Vet. Behav. 2020, 39, 21-31. doi: 10.1016/j.jveb.2020.06.002
Rizzo, M.; Arfuso, F.; Giannetto, C.; Giudice, E.; Longo, F.; Di Pietro, S.; Piccione, G. Cortisol levels and leukocyte population values in transported and exercised horses after acupuncture needle stimulation. J. Vet. Behav. 2017, 18, 56-61. doi.org/10.1016/j.jveb.2016.12.006
Rizzo, M.; Arfuso, F.; Giudice, E.; Abbate, F.; Longo, F.; Piccione, G. Core and surface temperature modification during road transport and physical exercise in horse after acupuncture needle stimulation. J Equine Vet. Sci. 2017, 55, 84–89. doi: 10.1016/j.jevs.2017.03.224
Fazio F. et al., Journal of Veterinary Behavior, 26, 5-10. The authors have reviewed this manuscript at the suggestion of the reviewer. The aim of this study was to evaluate parameters in ewes in response to handling and transport. Due to significant physiologic and transport practice differences the authors respectfully found this reference not to be pertinent to the objective of the current study. It has therefore not been included in the revised manuscript. However, a review paper by Ferlazzo, Fazio and Medica (2020) was found to one more directly relevant and has been inserted as a reference.
Carcangiu V. et al., Archives Animal Breeding, 61 (1), 37-41. The authors have reviewed this manuscript at the suggestion of the reviewer. The aim of this study was to evaluate milking ewe management. The authors respectfully found this reference not to be pertinent to the objective of the current study. It has therefore not been included in the revised manuscript.
Rizzo M., et al., Journal of Equine Veterinary Science 48 (2017) 23–30. The authors have reviewed this manuscript at the suggestion of the reviewer. The aim of this study was to evaluate acupuncture effects, rather that transport per se. The authors respectfully found this reference not to be sufficiently pertinent to the objective of the current study. It has therefore not been included in the revised manuscript.
Rizzo M., Arfuso F, Giannetto C, Giudice E, Longo F, Di Pietro S, Piccione G. (2017) Cortisol levels and leukocyte population values in transported and exercised horses after acupuncture needle stimulation. Journal of Veterinary Behavior, 18, 56-61. The authors have reviewed this manuscript at the suggestion of the reviewer. This article has now been cited in the manuscript.
Rizzo M., Arfuso F, Giudice E, Abbate F, Longo F, Piccione G. (2017) Core and surface temperature modification during road transport and physical exercise in horse after acupuncture needle stimulation. Journal of Equine Veterinary Science 55 (2017) 84–89. The authors have reviewed this manuscript at the suggestion of the reviewer. This article has now been cited in the manuscript.
Reviewer 2 Report
I would like to congratulate the authors on their submission. The paper itself is extremely well written and presented. The paper addresses an often-overlooked aspect of equine welfare and identifies a number of potential avenues for future research to further our understanding of how to prevent injuries during transportation. The methodology and analysis are clearly explained and the findings discussed in light of published literature in this area in the discussion.
I have a query about the data presented in Table 2 and a few very minor comments:
Line 78: and identifies (rather than ‘and to identify’)
Line 94: Delete Based], 1,055
Line 118: It would be useful to include the dates the survey was live here and to reiterate that it was capturing data from the two years preceding the date of completion so that the timeframe can be compared with other studies
Line 248: United Kingdom doesn’t need to be all in caps
Line 264: missing a full stop
Line 277: Interesting. Do you think that this difference may be due to differences in vehicle design and degree of access to other horses sharing the transport? Do you or the other study have the data to explore this?
Line 299: should you cite the citation number here rather than the year of Hall et al’s study?
Line 320: Is it possible that owners of horses who are prone to head injuries during transport are more likely to use poll protectors? These protectors also only cover a limited area of the head and may indeed provide protection to that area, but do not prevent injury elsewhere on the head
Figure 1: I’m not entirely sure what this image adds to the paper
Table 2: Section ‘A problem with the horse’ – please check these data. Loss of balance or falling (n=8) is repeated twice, although one is followed by (only) – it is unclear how these are different. Also the n listed in relation to each detail/example add up to 113 rather than 130.
Author Response
Dear Reviewer
Thank you for your feedback. We have addressed your feedback below and incorporated the changes into the revised manuscript.
Line 78: and identifies (rather than ‘and to identify’)
Change made as suggested - thanks
Line 94: Delete Based], 1,055
Change made as suggested - thanks
Line 118: It would be useful to include the dates the survey was live here and to reiterate that it was capturing data from the two years preceding the date of completion so that the timeframe can be compared with other studies
Dates added as suggested.
Line 248: United Kingdom doesn’t need to be all in caps
Change made as suggested - thanks
Line 264: missing a full stop
Thank you. I too noted this after submission (of course!)
Line 277: Interesting. Do you think that this difference may be due to differences in vehicle design and degree of access to other horses sharing the transport? Do you or the other study have the data to explore this?
Agreed. Given the preponderance of horse boxes on the UK compared with NZ, it may be associated with vehicle design. However, this may be confounded by different traffic conditions (e.g. the UK is far more populated) and driving practices. Unfortunately, we do not have sufficient granularity in our data to answer your question objectively. As always. More research needed!
Line 299: should you cite the citation number here rather than the year of Hall et al’s study?
Our understanding based on previous publications is that the normal convention when referencing an author directly by name is to give the year of publication in parentheses instead of the reference number. Happy to change is directed by the editor.
Line 320: Is it possible that owners of horses who are prone to head injuries during transport are more likely to use poll protectors? These protectors also only cover a limited area of the head and may indeed provide protection to that area, but do not prevent injury elsewhere on the head
Another can of worms I’m afraid. We have to be careful not to overinterpret the importance of a trend. However, when we examined transport-related behavioural problems in this population, we could not demonstrate a statistically significant association between using protective equipment and the perception of poor behaviour by respondents. It is a yet our unproven suspicion that owners of horses that are thought to be at risk of injury or poor behaviour are more likely use “protective equipment” To our knowledge there has been no specific study as to how best to design such equipment, and the poll protector certainly falls within this limitation.
Figure 1: I’m not entirely sure what this image adds to the paper
Noted. As scientists we are getting more pressure (or encouragement) to better engage the wider public in communicating our research. Infographics, short videos and other visual media are being requested by many journals now. Although the image portrays a real incident that may allow those unfamiliar with transport related injury to visualise some aspects of the problem, it also serves as a tool to engage a wider, perhaps less academic, audience. We hope that this rationale is acceptable.
Table 2: Section ‘A problem with the horse’ – please check these data. Loss of balance or falling (n=8) is repeated twice, although one is followed by (only) – it is unclear how these are different. Also the n listed in relation to each detail/example add up to 113 rather than 130.
The data used to generate Table 2 has been reorganised, and the table revised. Clarification between factors and subcategories within factors has now been added.